# A First Report on Side-Effects of COVID-19 Vaccines among General Population in Sudan: A Cross-Sectional Analysis

**DOI:** 10.3390/vaccines11020315

**Published:** 2023-01-31

**Authors:** Malik Suliman Mohamed, Ahmed Osman Mohamed, Rawaf Alenazy, Yusra Habib Khan, Mona Timan Idriss, Noura A. A. Alhudaib, Tilal Elsaman, Magdi Awadalla Mohamed, Eyman M. Eltayib, Tauqeer Hussain Mallhi

**Affiliations:** 1Department of Pharmaceutics, College of Pharmacy, Jouf University, Sakaka 72388, Saudi Arabia; 2Department of Pharmaceutics, Faculty of Pharmacy, University of Khartoum, Khartoum P.O. Box 1996, Sudan; 3Department of Pharmaceutical Microbiology, Faculty of Pharmacy, International University of Africa, Khartoum P.O. Box 2469, Sudan; 4Department of Medical Laboratory, College of Applied Medical Sciences-Shaqra, Shaqra University, Shaqra 11961, Saudi Arabia; 5Department of Clinical Pharmacy, College of Pharmacy, Jouf University, Sakaka 72388, Saudi Arabia; 6Department of Medical Sciences and Preparation Year, Northern College of Nursing, Arar 73312, Saudi Arabia; 7Department of Pharmaceutics, Faculty of Pharmacy, Imperial University College, Khartoum 11111, Sudan; 8Department of Pharmaceutical Chemistry, College of Pharmacy, Jouf University, Sakaka 72388, Saudi Arabia

**Keywords:** side effect, COVID-19, vaccine, Sudan, general population

## Abstract

Background: The process of mass immunization against COVID-19 may be impacted by vaccine reluctance despite intense and ongoing efforts to boost vaccine coverage. The COVID-19 vaccine is a crucial component for controlling the pandemic. To the best of our knowledge, we did not come across any study presenting the post-vaccination side-effect profile among the Sudanese population. Developing strategies to improve the vaccine acceptability and uptake necessitate evidence-based reports about vaccine’s side effects and acceptance. In this regard, this study aimed at estimating the prevalence of COVID-19 vaccine side-effects among the general population in Sudan. Methodology: A cross-sectional web-based quantitative study was conducted among the general population aged ≥18 years and residing in the Khartoum state of Sudan. A 30-item survey tool recorded the demographics, chronic diseases, allergy to other vaccines and COVID-19 vaccine side-effects after the first, second and booster doses. The data on the onset and duration of side-effects after each dose were also recorded. The distribution of side-effect scores after each dose of COVID-19 vaccine was compared using appropriate statistical methods. Results: A total of 626 participants were approached for this study. There was a preponderance of females (57.7%), and 19% of respondents had chronic diseases. The vaccination rate against COVID-19 was 55.8% (*n* = 349/626). The prevalence of side-effects after the first, second and booster doses were 79.7, 48 and 69.4%, respectively. Pain at the injection site, headache, fatigue, exhaustion and fever were the common side-effects after the first and second doses, while pain at the injection site, fatigue, headache and muscle pain were frequently reported after the booster dose. Most of these side-effects appeared within 6 h and resolved within one or two days following the administration of the vaccine dose. The average side-effects scores were 4.1 ± 4.4 (*n* = 349), 2.2 ± 3.6 (*n* = 202) and 3.5 ± 4.1 (*n* = 36) after the first, second and booster doses, respectively. The female gender had significantly higher side-effects after primary and booster doses. The age group 18-24 years indicated higher side-effects after the first dose compared to participants with ages ranging from 31 to 40 years (*p* = 0.014). Patients with chronic disease indicated significantly higher (*p* = 0.043) side-effects compared to those without any comorbid illness. Conclusions: This study showed a high prevalence of transient COVID-19 vaccine-related side-effects after primary and booster doses. However, these side-effects waned within 48 h. Pain at the injection site was the most common local side-effect, while fatigue, fever, headache and muscle pain were frequently reported systemic side-effects. The frequency of side-effects was more profound among females, young adults and those with comorbid conditions. These findings indicate that COVID-19 vaccines are safe and have side-effects as reported in the clinical trials of the vaccines. These results aid in addressing the ongoing challenges of vaccine hesitancy in the Sudanese population that is nurtured by widespread concerns over the safety profile.

## 1. Introduction

The coronavirus disease 2019 (COVID-19), which is brought on by the SARS-CoV-2, was declared to be a pandemic by the World Health Organization (WHO) on 11 March 2020 [1,2]. Since its discovery, SARS-CoV-2 has spread to about 200 countries, infected tens of millions of people and claimed the lives of over six million people [3,4]. To date, there is no effective therapy for COVID-19, leaving preventative measures such as mask use, hand washing and social isolation as the sole instant options to control viral transmission.

The highly contagious nature of the disease along with the high mortality rate demanded the swift development of vaccines against COVID-19. Vaccination seemed to be the most effective tool to control this pandemic [5]. The World Health Organization announced the release of various COVID-19 vaccinations in September 2020. Both the mRNA vaccine produced by Pfizer and the AstraZeneca (ChAdOx1 nCoV19) produced by Oxford received emergency use authorization (EUA) initially [6,7]. In Sudan, the first COVID-19 case was reported on 13 March 2020. Subsequently, Sudan received an invitation to participate in the COVID-19 vaccines global access (COVAX) facility which was accepted immediately [8]. It was the first country in the Middle East and North Africa (MENA) to receive AstraZeneca vaccines (initial 800,000 doses) under the COVAX facility. The Sudanese government initiated COVID-19 vaccination in priority groups, including healthcare professionals and high-risk populations. Initially, four vaccines against COVID-19 (Pfizer/BioNTech Comirnaty, Janssen (Johnson & Johnson, New Brunswick, NJ, USA) Jcovden, Oxford/AstraZeneca Vaxzevria, Sinopharm (Beijing, China) Covilo) were approved in Sudan [9]. However, various formulations of COVID-19 vaccines are currently for use in the general population [10].

According to a recent estimate, around 68% of the world’s population has received at least one dose of the COVID-19 vaccine with 13 billion doses administered globally at a rate of approximately 2 million per day [11]. As of 18 January 2023, Sudan has reported 63,717 COVID-19 cases along with 4998 deaths. A total of 13,711,970 doses have been administered, and 10,504,568 people have been vaccinated with at least one dose, while 8,302,878 (around 8.3 million) people have received full doses of COVID-19 vaccines [12]. In Sudan, vaccination against COVID-19 was initiated focusing on healthcare workers and those aged >45 years with comorbid conditions. It is important to mention that COVID-19 vaccines are not mandatory in Sudan. However, the ministry of health in Sudan encourages its population to take vaccines from nearby centers [13,14]. The low vaccination coverage in Sudan is quite alarming. Various national and international health authorities are struggling hard to boost vaccination coverage in the country [10]. In addition to the complicated logistics of vaccine production, testing, distribution and quality assurance, the acceptability of the COVID-19 vaccine is a significant global problem [15]. The COVID-19 vaccine is a crucial component in the battle against COVID-19. The process of mass immunization against COVID-19 may be impacted by vaccine reluctance despite intense and ongoing efforts to boost vaccine coverage [16]. Various studies have shown that the rapid development of COVID-19 vaccination along with uncertainty about the safety profile of the vaccines are the major factors that contribute towards the lack of trust in COVID-19 vaccination programs [13]. The safety profile of vaccines is still mandatory during the early phase of vaccination campaigns. The lack of data on the safety of vaccines may have a negative impact on vaccine uptake. Moreover, the safety data on the COVID-19 vaccines is not available in Sudan, that might be another factor linked with the vaccine hesitancy in the country. Available data indicate that the lack of trust in the safety and efficacy of COVID-19 vaccines is the major factor associated with vaccine hesitancy in Sudan [14,15]. To the best of our search, we did not come across any study presenting the post-vaccination side-effect profile among the Sudanese population. The sharing of data on the vaccines safety will aid to boost the confidence of the general population on vaccination campaigns, thereby increases the vaccine uptake. Moreover, real-world evidence will help the on the safety of vaccines will help in developing the strategies to improve the trust of the general public on vaccines. In this regard, this study aimed at estimating the prevalence of COVID-19 vaccine side-effects among the general population in Sudan.

## 2. Methodology

### 2.1. Ethical Approval

The ethical committee of the College of Pharmacy, International University of Africa approved this study and assigned the number: (Reference No.: 2022-cp324). In order to reduce the spread of COVID-19, the ethical committee approved the use of an online poll. All participants provided online consent following a brief explanation of the study. All data were subjected to anonymization before analysis.

### 2.2. Sample Size Estimation

As the vaccination coverage is unsatisfactory in Sudan, we did not estimate the sample size to include maximum population in this study. This kind of approach has been opted by several studies conducted during the COVID-19 pandemic [16,17].

### 2.3. Study Design, Setting and Population

A cross-sectional (March to June 2022) survey was disseminated among the general population from the Khartoum state of Sudan using Google Form. The participants were included in the current study if they were (1) adults aged ≥18 years, (2) received at least one dose of the COVID-19 vaccine, (3) residents of the Khartoum state and/or nearby suburbs, and (4) consented to participate in the current study.

### 2.4. Development and Validation of the Study Instrument

In order to identify the side-effects of COVID-19 vaccines, a thorough literature search was carried out [17,18], study instrument was developed and transferred to google form. The questionnaire was initially developed in English and translated into Arabic with at least 3 independent co-authors. To ensure that the questions’ original meaning was not altered, a second translation of the Arabic version into English was undertaken (forward and backward translation). The survey was rolled out in both English and Arabic so that the respondents may select their preferred version. The survey was initially validated through a team of experts from pharmacy and health colleges in Sudan and Saudi Arabia. The reliability of the questionnaire was assessed among 35 participants (pilot study). The correlations between study variables were checked and the reliability was confirmed (Coefficient alpha = 0.74).

The developed and validated study instrument (contains 3 sections) was used to collect the COVID-19 vaccines among participants. The first section collected information on demographics including age, gender, marital status, monthly income, education, smoking status, chronic diseases and use of medications for chronic diseases. Furthermore, participants were also asked whether they themselves or any member of their family got infected with COVID-19. The second section of the questionnaire gave a list of approved COVID-19 vaccinations in Sudan, and participants were asked to choose the vaccination which they received for their first vaccination dose, second vaccination dose and the booster dose. Any participant who did not get a second vaccination dose and/or booster dose was instructed to leave the column blank and chose the option of “I did not get a second dose of COVID-19 vaccination” and/or “I did not get a booster dose of COVID-19 vaccination”. The data of the onset and duration of side-effects in addition to other side-effect that was not included in the form were also collected. All the participants were asked to respond against each side-effect on a scale of “Yes” and “No”. The option “Yes” was scored “1”; otherwise, it scored “zero”. A cumulative side-effect score for each participant was estimated by adding the total number of side-effects.

### 2.5. Data Collection

The survey was transferred to Google form and distributed to the target population through convenient sampling. Initially, the survey was rolled out through various social media applications. The survey was made available for a specified study duration. The data was downloaded as Microsoft excel, cleaned based on exclusion/inclusion criteria, numerical codes assigned and then transferred to SPSP for analysis.

### 2.6. Statistical Analysis

The statistical package of social sciences (SPSS) version 25 was used for data analysis. Descriptive statistics were performed to generate a summary of the study variables as well as the responses of the participants towards the questionnaire items. Mean and standard deviation (SD) were used to present the continuous data, while categorical data were indicated as the frequency with proportions (%). Continuous data were compared using Student’s *t*-test or one-way ANOVA with Tukey’s post-hoc. For the purpose of analysis, participants were segregated into 3 groups on the basis of vaccination dose (single dose, double dose, booster dose). The side-effects experienced by participants after each type of vaccination dose were descriptively analyzed. The distribution of side-effects across demographic variables after each type of vaccination dose was analyzed. A significance level of *p* < 0.05 value was adjusted in all analyses.

## 3. Results

A total of 737 participants were approached, and 698 responded (response rate: 94.7%). Of these, 626 participants were included in the final analysis after applying the inclusion and exclusion criteria. There was a preponderance of female (57.7%) and graduate (64.9%) participants in the study population. Few participants (6.2%) had allergies to vaccines, and 19.3% of respondents had chronic diseases. Hypertension (46%), diabetes mellitus (5.4%) and asthma (5.4%) were the most prevalent comorbid conditions. About 19% of participants had a history of COVID-19 during the pandemic (Table 1).

### 3.1. Vaccination Rate and Side-Effects after the First Dose

The vaccination rate against COVID-19 was 55.8%. Of those who were vaccinated (*n* = 349), the majority received AstraZeneca (46.6%), followed by Janssen (30.4%) and Pfizer (20.1%) (Table 2). The prevalence of side-effects after the first dose was 79.7%. A total of 31 side-effects were reported by the study participants after the first dose. Pain at the injection site, headache, fatigue, exhaustion and fever were the common side-effects (Table 3). Most of these side-effects (35%) appeared within six hours after injection. Most of these side-effects resolved within one (25.8%) or two days (31.2%).

### 3.2. Side-Effects after the Second Dose

Of the 349 participants who received the first dose, 202 (57.9%) received the second dose of COVID-19 vaccines. Of these, the majority of the participants received AstraZeneca (69.3%), followed by Pfizer (23.3%) and Janssen (6.9%) (Table 4). The prevalence of side-effects after the second dose was 48%. A total of 23 side-effects were reported by the study participants after the second dose. Exhaustion, pain at the injection site, headache, fatigue and fever were the common side-effects (Table 5). Most of these side-effects (24.3%) appeared within six hours of vaccine administration. Most of these side-effects resolved within one (26.2%) to two days (18.3%).

### 3.3. Side-Effects after the Booster Dose

Of the 202 participants who received the second dose, only 36 (57.9%) received booster doses of COVID-19 vaccines. Of these, the majority of the participants received Moderna (41.7%), followed by Sinopharm (2.8%), AstraZeneca (27.8%) and Pfizer (27.8%) (Table 6). The prevalence of at least one side-effect after the booster dose was 69.4%. A total of 20 side-effects were reported by the study participants following the booster dose. Paint at the injection site, fatigue, headache and muscle pain, back pain and fever were the common side-effects (Table 7). Most of these side-effects (38.9%) appeared within six hours after the injection. Most of these side-effects resolved within one (22.2%) to two days (27.8%).

### 3.4. Distribution of Side-Effect Scores among Demographic Characteristics

The side-effect score (SES) was estimated after each dose of COVID-19 vaccine. The average side-effects scores were 4.1 ± 4.4 (*n* = 349), 2.2 ± 3.6 (*n* = 202) and 3.5 ± 4.1 (*n* = 36) after the first, second and booster doses, respectively. The female gender had significantly higher SES after the primary and booster doses of COVID-19 vaccines. The age group 18–24 years indicated a higher SES after the first dose compared to participants with ages ranging from 31 to 40 years (*p* = 0.014). However, the SES was equally distributed among age groups after the second and booster doses. Additionally, the SES was higher among widows after both primary doses. Patients with chronic disease indicated significantly higher (*p* = 0.043) SES compared to those without any comorbid illness. The participants who reported to have an allergy with any vaccine indicated a higher SES after the booster dose. However, the SES was higher among vaccine-allergic participants after the first and second dose, but the differences were not statistically significant. All other demographics, including the types of COVID-19 vaccines, indicated the equal distribution of SES (Table 8).

## 4. Discussion

To the best of our knowledge, this is the first report on the safety profile of COVID-19 vaccines in Sudan. The low vaccine acceptance in Sudan is a major barrier to implementing a successful vaccination campaign [15,19,20]. The available literature underscores that the fear of unknown side-effects is substantially linked with the low acceptability of COVID-19 vaccines in Sudan [14,15]. In this context, it is imperative to inform the general public regarding real-world data on the safety of COVID-19 vaccines. Our findings indicate a low vaccination rate among study participants. More than 50% of the study population experienced at least one side-effect following the primary and booster doses of COVID-19 vaccines. Pain at the injection, fatigue, headache, fever and muscle pain were frequently reported side-effects after the first, second and booster doses. Most of these side-effects occurred within 6 h of vaccine administration and resolved within 48 h following the doses. Female gender, young participants, respondents with an allergy to any vaccine and those with underlying comorbid conditions indicated higher side-effects scores. However, the types of vaccines were not linked with the frequency of side-effects. Nevertheless, all of the reported side-effects were minor and transient and did not result in any major adverse event among vaccines. Taken together, the benefits of COVID-19 vaccines outweigh the risks among the Sudanese population. The findings of this study will assist the health authorities to curb vaccine hesitancy due to the side-effects of COVID-19 vaccines in the Sudanese population.

The prevalence of at least one side-effect ranged from 48 to 80% in this study, where the highest proportion of side-effects was reported after the first dose. These results are in concordance with other studies [17,18,19]. A study from another African country, such as Ghana, reported an 81% rate of at least one side-effect following the administration of the AstraZeneca vaccine [20]. Another study conducted in an Arabic region (Saudi Arabia) reported the incidence of at least one side-effect as 88.7 and 95.1% after the first and second doses of COVID-19 vaccines, respectively [18]. However, the incidence of side-effects after the second dose was comparatively lower (48%) in our study. Such wide variations in the reported frequencies might be attributed to the study population and type of vaccines. Mallhi et al. conducted a study on people with underlying diseases where the use of Pfizer vaccines was more profound [18]. Another study from Ethiopia indicated that around half of the study participants who had received AstraZeneca COVID-19 vaccines experienced at least one local and systematic side-effect [21]. The inconsistency of these results with our findings might be attributed to the disparities in the study population i.e., healthcare professionals vs. the general population. The high prevalence of side-effects in our study compared to the other studies can be explained by the use of different vaccines. More than two-thirds of the study population in our study received Oxford-AstraZeneca. The existing evidence suggests a higher frequency of side-effects among Oxford-AstraZeneca recipients compared to recipients of the Sinovac and Pfizer vaccines [22]. The higher proportion of side-effects after the first dose of COVID-19 vaccines in our study is aligned with other studies [23,24,25]. The variations in the incidence of at least one side-effect across the studies can be explained by various factors, including study population, type of vaccine and participants’ demographics. Our analysis confirms the high prevalence of side-effects after the COVID-19 vaccine. However, these local and systematic effects were transient and short-lived, and all have already been indicated by the clinical trials on these vaccines. There is a need for more studies to ascertain the primary covariate affecting the incidence of side-effects following the administration of COVID-19 vaccines.

The distribution of side-effects in our study after the primary and booster doses of COVID-19 vaccines corroborates with the findings of other studies [17,18,26,27]. The most common local side-effect reported in this study was pain at the injection site, while fatigue, headache, fever and muscle pain were frequently reported systemic side-effects among vaccines. It is interesting to note that most of these side-effects appeared within 6 h of vaccine administration and waned within 48 h. These results are consistent with the findings of Mallhi et al. [18]. Yesuf et al. reported the waning of side-effects within 72 h, and this prolonged duration might be associated with the study population [21], as healthcare professionals perform more vigilant monitoring of side-effects than the general population. It might be possible that people considered alleviating the severity of side-effects as a resolving point in our study. All the side-effects reported in our study were self-limited and seemed to occur due to the provocation of the immune system due to the administration of the vaccines. The sharing of these results with the general population will boost confidence in the vaccines and accelerate vaccination coverage in Sudan.

It is pertinent to mention that the side-effect profile in this study was self-reported. In this context, several covariates must be considered when interpreting the results. Our analysis found that various demographic features were linked with the high SES. For example, the females reported more side-effects than males. These findings are in agreement with other studies [17,22,28]. Females have stronger antibody, innate and adaptive immune responses compared to males [29]. The higher frequency of side-effects among females can be explained by these biological mechanisms. Other studies have reported higher side-effects among males compared to females [30]. It is important to note that the perceived severity of side-effects may vary across gender, and there is a need for further investigations to establish the relationship between gender and the frequency of side-effects. Hatmal et al. reported a higher frequency of pain at the injection site and tiredness after COVID-19 vaccines among females compared to males [31]. This reporting pattern is quite reasonable due to the high sensitivity and lower thresholds for pain among females. Males and females tend to react differently to COVID-19 vaccines due to their hormonal homeostasis and genetic makeup [32]. Recent investigations have confirmed the findings that the administration of Pfizer-BioNTech, AstraZeneca, Sinopharm, Sputnik V, SinoVac, Johnson & Johnson and Moderna vaccines result in a higher frequency of side-effects in females compared to males [31]. The other factor indicating a higher frequency of side-effects was the age of the study participants. Young people reported more side-effects after the first dose of COVID-19 vaccines compared to the other age groups included in this study. These results are aligned with the findings of Elnaeam et al., where the authors reported that the younger age group (18–30 years) had 7.4 times higher odds to experience vaccine-related side-effects [22]. It is worth mentioning that the relationship between age and the frequency of side-effects was not observed after the second and booster doses, and the small sample size might be a possible reason for such findings. Our analysis also observed a higher side-effect score among people with chronic conditions. The relationship between chronic diseases with the frequency and severity of side-effects among vaccine recipients has been established [18]. These findings are in concordance with another study from the United Arab Emirates (UAE), where study participants with comorbid conditions had higher number of side effects as compared to those without any underlying illnesses [33]. Other studies conducted in Arab regions have also indicated a substantial association of chronic diseases with the development [34], as well as frequency and severity, of post vaccination side-effects [31]. However, this association has been negated in other studies. A study among Turkish healthcare workers who had received Sinovac vaccine reported no association of chronic diseases with the occurrence and severity of the side-effects [35]. The participants whose marital status was “widow” indicated the highest SES after the first and second doses of COVID-19 vaccines. However, the post-hoc analysis indicated that the participants who were single had significantly higher SES after the first dose than those who were married. On the other hand, widowers had significantly higher SES than single and married participants. Similar findings have been observed in another study, where the participants with divorced status experienced higher side-effects compared to single or married participants [17]. The relationship between marital status and the frequency and severity of side-effects is complex and may relate to several factors, including psychological well-being, health-relevant immune alterations and depressogenic perpetuators linked to the marital status of the individuals [36]. However, further investigations are required to establish such a relationship. Only 6.2% of the participants in our study had a previous history of allergies to any vaccine. The SES scores were higher among those with a history of vaccine allergy after the first and second doses, but the difference was statistically insignificant. However, the SES was significantly higher among participants with a history of vaccine allergy after receiving the booster doses. The relationship between previous vaccine allergies and the frequency of side-effects cannot be established with the small number of participants in our study. The previous research has found that participants who have had allergic reactions to any vaccine may have allergic reactions to COVID-19 vaccines [37]. However, the higher frequency of side-effects among people who had a history of vaccine allergies might be linked with high perceived risks of side-effects among individuals due to their previous experience with the vaccines. In this context, the reporting bias should not be disregarded when interpreting the results. Nevertheless, these findings warrant further investigations to establish the relationship between host previous immune sensitization with vaccines and the occurrence of side-effects following COVID-19 vaccines. Our results urge a need for the vigilant monitoring of side-effects according to the demographic features of vaccine recipients.

It is important to mention that the side-effects following COVID-19 vaccines are similar to those reported with other vaccines that have been in use for decades [38,39,40,41]. Most of these side-effects are attributed to the provocation or sensitization of the immune system following the vaccination. In this context, our findings aid to boost the confidence of the general population towards COVID-19 vaccination amid its comparable safety profile to other vaccines already in use.

Unfortunately, Sudan is experiencing several challenges in implementing a successful COVID-19 vaccination program. Despite continuous support from COVAX, the vaccination coverage in the country is not satisfactory. According to a recent estimate, around 20% of the country’s population has been fully vaccinated, which is far below the global target [11]. Severe factors, including a descending economy, political turmoil, lack of knowledge about vaccines, lack of trust in health authorities, conspiracy beliefs, vaccine inequity, perceived dangers of vaccine-related side-effects and limited resources contribute to poor vaccination coverage in Sudan. Although the current study was conducted in the Khartoum region, where the health facilities and their access are somewhat better than in other parts of the country, the vaccination rate is still alarmingly low, and only a few participants have received the second dose, even after a considerable time has elapsed. Our findings also serve as a call to action for national and international health regulatory authorities to implement multilateral approaches to improve vaccination coverage. Informing the general population regarding the safety of COVID-19 vaccines will neutralize negative beliefs regarding the vaccines among the general population.

The findings of the current study are accompanied by a few shortcomings that should be considered when interpreting the results. The small sample size is a major limitation of this study, as a large sample size may indicate different results with more statistical power. Information, recall and reporting bias are noticeable demerits of this study. Moreover, the convenient sampling further adds to the selection bias. This study did not evaluate the long-term impact of side-effects, thereby necessitating the need for longitudinal surveys. As a large sample size may result in variable inferences, future studies should consider a larger cohort to verify and replicate our findings. Additionally, as the participants were recruited from Khartoum and nearby suburbs where the health facilities are somewhat better than in other regions of the country, the findings may not be generalized for the poorly developed or developing regions of Sudan. Despite these limitations, our findings are strengthened by a first report on the side-effect profile of COVID-19 vaccines among the Sudanese population. The findings of the current study will help instill confidence in COVID-19 vaccines and, subsequently, increase vaccine coverage in this population. Nevertheless, this study confirms the safety of COVID-19 vaccines, and its findings can be utilized in creating awareness among the general population in Sudan.

## 5. Conclusions

Our analysis showed mild and transient side effects following the primary and booster doses of COVID-19 vaccines. The side effects reported in this study were comparable to the findings of phase 3 clinical trials. However, these side-effects were short-lived and waned within 48 h. Pain at the injection site was frequently reported local side-effect, while fatigue, fever, headache and muscle pain were commonly reported systemic side-effects. The frequency of side-effects was more profound among females, young adults, and those with comorbid conditions and a history of allergy with any vaccine. However, the type of vaccines did not show any significant difference for the frequency of side-effects. The side-effects reported in this study were comparable to the results of other studies and findings from clinical trials of the vaccines. It is imperative to mention that these results should be validated, verified, and replicated through active pharmacovigilance or qualitative studies. These findings may aid in addressing the ongoing challenges of vaccine hesitancy in the Sudanese population that is nurtured by widespread concerns over safety profiles.

## Figures and Tables

**Table 1 vaccines-11-00315-t001:** Participants’ demographics (N = 626).

Variables	Frequency (N)	Percentage (%)
*Gender*		
Female	361	57.7%
Male	265	42.3%
*Age in* years		
18–24	258	41.2%
25–30	148	23.6%
31–40	83	13.3%
41–50	76	12.1%
>50	61	9.7
*Marital status*		
Married	383	61.2%
Single	225	35.9%
Divorced	10	1.6%
Widow	8	1.3%
*Monthly income (Sudanese pound SDG)*		
<50,000 thousand	266	42.5%
50–100 thousand	189	30.2%
100–200 thousand	91	14.5%
200–400 thousand	50	8.0%
>400 thousand	30	4.8%
*Education level*		
Khalwa	10	1.6%
Intermediate or elementary	12	1.9%
Secondary	47	7.5%
Graduate	406	64.9%
Postgraduate	151	24.1%
*Residence*		
Khartoum	450	71.9%
Omdurman	106	16.9%
Bahri	70	11.2%
*Smoking status*		
Non-smoker	533	85.1%
Current smoker	64	10.2%
Ex-smoker	29	4.6%
*Do you have allergy with any vaccine?*		
Yes	39	6.2%
No	587	93.8%
*Do you have any Chronic disease?*		
Yes	121	19.3%
No	505	80.7%
*Types of chronic disease*		
Hypertension (high blood pressure)	46	7.3%
Diabetes	34	5.4%
Asthma	33	5.3%
High cholesterol	10	1.6%
Allergy	8	1.3%
Cardiovascular diseases	4	0.6%
Irritable bowel syndrome	4	0.6%
Kidney diseases	2	0.3%
Anemia	2	0.3%
Thyroid disorder	2	0.3%
Multiple sclerosis	1	0.2%
Gout	1	0.2%
Rheumatoid arthritis	1	0.2%
Migraine	1	0.2%
*Are you using medication for any chronic disease?*		
Yes	92	14.7%
No	534	85.3%
*Did you get COVID-19 infection during current pandemic?*		
Yes	117	18.7%
No	509	81.3%
*Did anyone in your family get COVID-19 infection during the current pandemic?*		
Yes	230	36.7%
No	396	63.3%

**Table 2 vaccines-11-00315-t002:** Participants receiving the first dose of COVID-19 vaccines.

Variables	Frequency (N)	Percentage (%)
*Did you get the FIRST Dose of COVID-19 Vaccine? (N = 626)*		
Yes	349	55.8%
No	277	44.2%
*Which of the following COVID-19 vaccine have you received as FIRST DOSE? (N = 349)*		
Oxford-AstraZeneca	162	46.4%
Janssen (Johnson & Johnson)	106	30.4%
Pfizer-BioNTech	70	20.1%
Moderna	5	1.4%
Sinopharm	6	1.7%
Prevalence of side-effects after the first dose		
Experienced at least one side-effect	278	79.7%
Did not experience any side-effects	71	20.3%
Onset of side-effects		
Immediately	40	11.5%
Within 6 h after injection	122	35.0%
Within 12 h after injection	67	19.2%
Within 12–24 h after injection	31	8.9%
After 24 h of injection	18	5.2%
Duration of side-effects		
Less than 24 h	90	25.8%
2 days	109	31.2%
3 days	43	12.3%
4 days	12	3.4%
5 days	12	3.4%
Greater than 5 days	12	3.4%

**Table 3 vaccines-11-00315-t003:** Side-effects experienced after first dose of COVID-19 vaccines (N = 349).

Variables	Frequency (N)	Percentage (%)
Pain at the injection site	176	50.4%
Fatigue	175	50.1%
Headache	137	39.3%
Fever	128	36.7%
Feeling unwell	115	33.0%
Muscle pain	88	25.2%
Generalized body pain	81	23.2%
Swelling at the injection site	57	16.3%
Back pain	55	15.8%
Redness at the injection site	53	15.2%
Itching at injection site	46	13.2%
Flu	39	11.2%
Chills	36	10.3%
Sore throat	31	8.9%
Nausea or vomiting	24	6.9%
Shortness of breath	22	6.3%
Diarrhea	20	5.7%
Loss of smell	20	5.7%
Loss of taste	16	4.6%
Cough	12	3.4%
Insomnia	3	0.9%
Dizziness	2	0.6%
Epilepsy	1	0.3%
Menstrual cycle irregularities	1	0.3%
Loss of appetite	1	0.3%
Drowsiness	1	0.3%
Rapid heart beat	1	0.3%
Numbness	1	0.3%
Hallucination	1	0.3%

**Table 4 vaccines-11-00315-t004:** Participants receiving the second dose of COVID-19 vaccines.

Variables	Frequency (N)	Percentage (%)
Did you get the Second Dose of COVID-19 Vaccine? (N = 349)		
No	147	42.1%
Yes	202	57.9%
Which of the following COVID-19 vaccines have you received as Second DOSE? (N = 202)		
Oxford-AstraZeneca	129	63.9%
Janssen (Johnson & Johnson)	14	6.9%
Pfizer-BioNTech	47	23.3%
Moderna	8	4.0%
Sinopharm	4	2.0%
Prevalence of side-effects after the second dose		
Experienced at least one side-effect	97	48%
Did not experience any side-effects	105	52%
Onset of side-effects		
Immediately	14	6.9%
Within 6 h after injection	49	24.3%
Within 12 h after injection	23	11.4%
Within 12–24 h after injection	14	6.9%
After 24 h of injection	11	5.4%
*Duration of side-effects*		
Less than 24 h	53	26.2%
2 days	37	18.3%
3 days	15	7.4%
5 days	3	1.5%
Greater than 5 days	3	1.5%
*Duration between first and second dose*		
2 Weeks	9	4.5%
3 Weeks	12	5.9%
4 Weeks	36	17.8%
6 Weeks	17	8.4%
8 Weeks	51	25.2%
10 Weeks	20	9.9%
12 Weeks	22	10.9%
Greater than 12 Weeks	35	17.3%

**Table 5 vaccines-11-00315-t005:** Side-effects experienced after second dose of COVID-19 vaccines (N = 202).

Variables	Frequency (N)	Percentage (%)
Fatigue	134	66.3%
Pain at the injection site	45	22.3%
Headache	44	21.8%
Fever	38	18.8%
Feeling unwell	30	14.9%
Generalized body pain	29	14.4%
Muscle pain	28	13.9%
Back pain	18	8.9%
Redness at the injection site	16	7.9%
Sore throat	12	5.9%
Swelling at the injection site	12	5.9%
Chills	11	5.4%
Itching at injection site	9	4.5%
Flu	9	4.5%
Shortness of breath	9	4.5%
Diarrhea	9	4.5%
Nausea or vomiting	8	4.0%
Loss of smell	6	3.0%
Loss of taste	4	2.0%
Cough	3	1.5%
Loss of voice	1	0.5%
Hand pain	1	0.5%
Drowsiness	1	0.5%

**Table 6 vaccines-11-00315-t006:** Participants receiving the booster dose of COVID-19 vaccines.

Variables	Frequency (N)	Percentage (%)
Did you get the Booster Dose of COVID-19 Vaccine? (N = 202)		
No	166	82.2%
Yes	36	17.8%
Which of the following COVID-19 vaccine have you received as Booster DOSE? (N = 36)		
Oxford-AstraZeneca	10	27.8%
Pfizer-BioNTech	10	27.8%
Moderna	15	41.7%
Sinopharm	1	2.8%
Prevalence of side-effects after the booster dose		
Experienced at least one side-effect	25	69.4%
Did not experience any side-effects	11	30.6%
Onset of side-effects		
Immediately	2	5.6%
Within 6 h after injection	14	38.9%
Within 12 h after injection	4	11.1%
Within 12–24 h after injection	3	8.3%
After 24 h of injection	2	5.6%
Duration of side-effects		
Less than 24 h	8	22.2%
2 days	10	27.8
3 days	5	13.9%
4 days	1	2.8%
Greater than 5 days	1	2.8%
Duration between second and booster dose		
1 Month	3	8.3%
2 Month	3	8.3%
3 Month	6	16.7%
4 Month	7	19.4%
5 Month	2	5.6%
6 Month	7	19.4%
Greater than 6 Months	4	11.1%
Not sure	4	11.1%

**Table 7 vaccines-11-00315-t007:** Side-effects experienced after booster dose of COVID-19 vaccine (N = 36).

Variables	Frequency (N)	Percentage (%)
Fatigue	16	44.4%
Pain at the injection site	15	41.7%
Headache	13	36.1%
Muscle pain	11	30.6%
Back pain	11	30.6%
Fever	11	30.6%
Feeling unwell	8	22.2%
Generalized body pain	6	16.7%
Chills	4	11.1%
Swelling at the injection site	3	8.3%
Sore throat	3	8.3%
Flu	3	8.3%
Diarrhea	3	8.3%
Itching at injection site	2	5.6%
Nausea or vomiting	2	5.6%
Cough	2	5.6%
Loss of smell	2	5.6%
Loss of taste	1	2.8%
Shortness of breath	1	2.8%
Drowsiness	1	2.8%

**Table 8 vaccines-11-00315-t008:** Distribution of side-effect score among demographic features after primary and booster doses of COVID-19 vaccines.

Variables	First Dose	Second Dose	Booster Dose
Gender			
Female	4.8 (4.57)	2.74 (4.34)	5.24 (4.7)
Male	3.35 (4.02)	1.51 (2.5)	1.95 (2.86)
*p*-value	0.002	0.014	0.019
Age in years			
18–24	4.98 (4.76)	2.46 (4.3)	6.75 (7.89)
25–30	4.35 (4.20)	2.57 (3.8)	5.33 (476)
31–40	2.7 (3.16)	1.36 (2.45)	3.25 (2.34)
41–50	3.25 (3.51)	2.19 (3.21)	3.2 (3.94)
>50	4.33 (5.48)	2.12 (4.07)	1.13 (1.13)
*p*-value	0.014	0.53	0.17
Post-hoc analysis	18–24 versus 31–40 = 0.014		
Marital status			
Single	4.68 (4.5)	2.3 (3.72)	5.5 (5.54)
Married	3.26 (4.01)	1.79 (3.13)	2.72 (3.32)
Divorced	5.25 (2.63)	4.0 (5.66)	3.0
Widow	5.83 (6.43)	10.0 (11.13)	-
*p*-value	0.017	0.011	0.2
*p*-value with post-hoc tests	Single versus married = 0.015	Single vs. widow = 0.014 Married vs. widow = 0.008	
Monthly income (Sudanese pound SDG)			
<50,000	4.06 (4.3)	2.68 (4.56)	5.44 (5.41)
50–100 thousand	4.24 (4.4)	1.47 (2.41)	1.20 (2.17)
100–200 thousand	4.12 (4.0)	1.98 (2.84)	3.86 (3.85)
200–400 thousand	4.03 (5.46)	2.67 (4.45)	3.0 (3.22)
>400 thousand	3.74 (3.93)	2.35 (3.73)	2.89 (4.14)
*p*-value	0.99	0.41	0.44
*Education level*			
Intermediate or elementary	3.67 (4.48)	1.43 (1.4)	.
Secondary	4.29 (5.35)	2.55 (4.25)	1.25 (1.0)
Graduate	4.35 (4.6)	2.03 (3.78)	3.44 (4.42)
Postgraduate	3.62 (3.5)	2.32 (3.32)	4.21 (4.25)
*p*-value	0.54	0.85	0.46
*Residence*			
Khartoum	4.13 (4.59)	2.38 (3.89)	3.72 (4.33)
Omdurman	3.83 (3.25)	1.26 (2.35)	4.33 (4.51)
Bahri	4.28 (4.15)	1.67 (2.5)	1.25 (1.5)
*p*-value	0.88	0.25	0.51
*Smoking status*			
Non-smoker	4.09 (4.33)	2.29 (3.81)	4.03 (4.35)
Current smoker	4.73 (5.16)	1.57 (2.77)	1.33 (2.16)
Ex-smoker	2.67 (2.26)	1.38 (1.71)	1.0
*p*-value	0.30	0.51	0.29
*Do you have allergy with any vaccine?*			
No	4.01 (4.31)	2.08 (3.51)	3.15 (3.92)
Yes	5.82 (5.22)	3.67 (5.48)	9.5 (3.54)
*p*-value	0.1	0.2	0.03
Do you have any Chronic disease?			
No	3.8 (4.0)	2.03 (3.35)	3.68 (4.03)
Yes	5.19 (5.46)	2.59 (4.44)	3.09 (4.53)
*p*-value	0.043	0.36	0.7
Types of chronic disease			
Hypertension			
No	4.02 (4.28)	2.11 (3.69)	3.55 (4.34)
Yes	4.9 (5.19)	2.5 (3.07)	3.2 (2.77)
*p*-value	0.29	0.64	0.86
Diabetes			
No	4.11 (4.32)	2.15 (3.51)	3.97 (4.34)
Yes	3.96 (4.97)	2.17 (4.72)	1.57 (2.51)
*p*-value	0.87	0.99	0.17
High cholesterol			
No	4.1 (4.37)	2.17 (3.65)	3.5 (4.13)
Yes	4.0 (4.53)	1.5 (1.73)	-
*p*-value	0.96	0.72	-
Kidney diseases			
No	4.07 (4.36)	2.16 (3.64)	3.5 (4.13)
Yes	8.5 (4.95)	2.0 (0)	-
*p*-value	0.15	0.95	-
Asthma			
No	4.02 (4.34)	2.15 (3.66)	3.5 (4.13)
Yes	5.73 (4.76)	2.33 (2.07)	-
*p*-value	0.41	0.9	-
Irritable bowel syndrome			
No	4.11 (4.38)	2.16 (3.62)	3.5 (4.13)
Yes	2.5 (2.12)	-	-
*p*-value	0.61	-	-
Thyroid disorder			
No	4.078 (4.36)	2.15 (3.63)	3.5 (4.13)
Yes	7.5 (6.36)	2.0	-
*p*-value	0.27		
Are you using medication for any chronic disease?			
No	3.91 (4.11)	1.95 (3.27)	3.6 (4.06)
Yes	4.98 (5.36)	3.05 (4.79)	3.27 (4.47)
*p*-value	0.14	0.09	0.83
Did you get COVID-19 infection during current pandemic?			
No	3.93 (4.24)	1.97 (3.5)	3.53 (4.38)
Yes	4.64 (4.74)	2.57 (3.88)	3.47 (3.97)
*p*-value	0.20	0.28	0.97
Did anyone in your family get COVID-19 infection during the current pandemic?			
No	3.79 (4.15)	1.92 (3.4)	2.69 (3.52)
Yes	4.52 (4.62)	2.42 (3.85)	4.15 (4.55)
*p*-value	0.12	0.33	0.3
Did anyone in your friends or relatives get COVID-19 infection during the current pandemic?			
No	3.63 (4.2)	2.79 (4.17)	2.67 (4.68)
Yes	4.26 (4.42)	1.98 (3.45)	3.67 (4.08)
*p*-value	0.24	0.19	0.6
COVID-19 vaccines			
Oxford-AstraZeneca	4.8 (4.9)	1.9 (3.7)	4.1 (5.3)
Janssen (Johnson & Johnson)	3.5 (3.6)	1.9 (3.0)	-
Pfizer-BioNTech	3.7 (4.1)	2.4 (3.1)	2.3 (3.4)
Moderna	1.8 (1.1)	5.3 (6.0)	4.1 (3.9)
Sinopharm	2.3 (2.2)	1.3 (1.5)	-
*p*-value	0.051	0.131	0.565

## Data Availability

Authors will share the paper in both social media and Universities websites where they are working, in addition to publicly archived datasets or others that suggested by MDPI.

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
