# Peer review of "A First Report on Side-Effects of COVID-19 Vaccines among General Population in Sudan: A Cross-Sectional Analysis"

_vaccines, 2023, doi:10.3390/vaccines11020315_

Round 1

Reviewer 1 Report

The manuscript by Mohamed et al entitled A First Report on Side Effects of COVID-19 Vaccines among 2 General Population in Sudan: A Cross-sectional analysis, surveyed 626 subjects in Sudan (349 vaccinated) to determine the reported side-effects of COVID-19 vaccinations. The paper concludes that there was a high prevalence of side-effects, but these were minor and short-lived.

Comments

Reference 2 needs to be updated. This reference is from 2020 and already quite out of data in respect to mortality data. Even in 2020 WHO estimated around 3M excess deaths (https://www.who.int/data/stories/the-true-death-toll-of-covid-19-estimating-global-excess-mortality).

The current estimates are well over 6M deaths.

The introduction states “As of December 15, 2022, Su-81 dan has reported 63,663 COVID-19 cases along with 4,992 deaths”. This statement needs some qualification as to the reliability of this figure. The number of deaths represents 7.8% of the total cases. This is unusually high, and most estimates put the death rate under 3% of infections. Also the total number of infections reported here (63,663) in a country with a population of around 45M seems extremely low.

The introduction mentions the concept of ‘herd immunity’ and proposes that this is “a viable weapon against the COVID-19 pandemic”. The idea of herd immunity has faced significant challenges and is now not a serious proposition due to several reasons including the lack of sterilizing or transmission-blocking immunity due to vaccination or previous infection, decline in immunity over time and the appearance of new viral variants.

The types of side effect described e.g. pain at the injection site, fatigue, muscle pain, headaches and chills are also commonly seen with other vaccines. The discussion could include some information about this. This would put COVID-19 vaccines in context and show the public that the main side effects are comparable to other vaccines already in use.

The main limitations of the study should be highlighted (e.g. small population size)

Minor comments

The Discussion states “Taken together, the high prevalence of side effects in our study can be linked to the type of vaccine”. This type of sentence could be taken out of context by certain media outlets. Given that all side effects reported are minor it might be better to say “Taken together, the high prevalence of minor and transient side effects in our study can be linked to the type of vaccine”

The same applies to the first sentence of the conclusion “Our analysis showed a high prevalence of COVID-19 vaccine-related side effects after primary and booster doses”. Since the public is not going to read or understand the paper, the type of sentences in the conclusion will be cited by the media, commentators or politicians and should be carefully written. Writing “Our analysis showed a high prevalence of COVID-19 vaccine-related side effects” may not be what the authors want to convey to the public.

Please use standard IM abbreviations in the references section.

Author Response

Point-by-Point response to reviewers

Respected Editor,

We have received comments for our manuscript. All the concerns and suggestions of the reviewers have been addressed and we hope the revised version of the manuscript will satisfy the concerns of all reviewers. We have attached a point-by-point response to each reviewer and revised version of the manuscript for your consideration. Please let us know if any other changes are required in this regard. Last but not least, we are very thankful to the editor and all reviewers for their time and efforts to put valuable suggestions. Indeed, their recommendations made this manuscript more scientifically elegant and sound.

Query 1: "The manuscript by Mohamed et al entitled A First Report on Side Effects of COVID-19 Vaccines among 2 General Population in Sudan: A Cross-sectional analysis, surveyed 626 subjects in Sudan (349 vaccinated) to determine the reported side-effects of COVID-19 vaccinations. The paper concludes that there was a high prevalence of side-effects, but these were minor and short-lived.

Query reply 1: Respected reviewer, thank you very much for summarizing and acknowledging our work. Considering the need of such studies in Sudan, we tried our best to provide a real-world data on the vaccine safety in the country. We have incorporated the suggested changes in the revised version of the manuscript.

Comments

Query 2: Reference 2 needs to be updated. This reference is from 2020 and already quite out of data in respect to mortality data. Even in 2020 WHO estimated around 3M excess deaths (https://www.who.int/data/stories/the-true-death-toll-of-covid-19-estimating-global-excess-mortality). The current estimates are well over 6M deaths.

Query reply 2: Respected reviewer, we agree with you suggestion and have updated the reference to https://covid19.who.int/ . Moreover, we have also replaced “over a million deaths” with “over six million deaths”. Thank you for this constructive comment.

Query 3: The introduction states “As of December 15, 2022, Su-81 dan has reported 63,663 COVID-19 cases along with 4,992 deaths”. This statement needs some qualification as to the reliability of this figure. The number of deaths represents 7.8% of the total cases. This is unusually high, and most estimates put the death rate under 3% of infections. Also the total number of infections reported here (63,663) in a country with a population of around 45M seems extremely low.

Query reply 3: Respected reviewer, we have updated the reference for these figures. We agree with your concern, but now the reference has been provided from World Health Organization.

Query 4: The introduction mentions the concept of ‘herd immunity’ and proposes that this is “a viable weapon against the COVID-19 pandemic”. The idea of herd immunity has faced significant challenges and is now not a serious proposition due to several reasons including the lack of sterilizing or transmission-blocking immunity due to vaccination or previous infection, decline in immunity over time and the appearance of new viral variants.

Query reply 4: Respected reviewer, thank you very much for this valuable information. We have rephrased the sentences where the discussion was made in context of herd immunity. Now our manuscript does not contain information on vaccination related herd immunity.

Query 5: The types of side effect described e.g. pain at the injection site, fatigue, muscle pain, headaches and chills are also commonly seen with other vaccines. The discussion could include some information about this. This would put COVID-19 vaccines in context and show the public that the main side effects are comparable to other vaccines already in use.

Query reply 5: Respected reviewer, thank you very much for this suggestion. Indeed, it is a valuable addition to our manuscript. We have added a passage (3rd last paragraph) in the discussion section where we acknowledge that the covid-19 vaccines have comparable safety profile to other vaccines already in use.

Query 6: The main limitations of the study should be highlighted (e.g., small

population size)

Query reply 6: Respected reviewer, we have added the limitation of sample size in the manuscript.

Minor comments

Query 7: The Discussion states “Taken together, the high prevalence of side effects in our study can be linked to the type of vaccine”. This type of sentence could be taken out of context by certain media outlets. Given that all side effects reported are minor it might be better to say “Taken together, the high prevalence of minor and transient side effects in our study can be linked to the type of vaccine”

Query reply 7: Respected reviewer, we agree with your suggestion and have modified the sentence at the end of first paragraph of discussion. I hope that the revised version will satisfy the concerns.

Query 8: The same applies to the first sentence of the conclusion “Our analysis showed a high prevalence of COVID-19 vaccine-related side effects after primary and booster doses”. Since the public is not going to read or understand the paper, the type of sentences in the conclusion will be cited by the media, commentators or politicians and should be carefully written. Writing “Our analysis showed a high prevalence of COVID-19 vaccine-related side effects” may not be what the authors want to convey to the public.

Query reply 8: Respected reviewer, we agree with your suggestion and have modified the first sentence of the conclusion section. I hope that the revised version will satisfy the concerns.

Query 9: Please use standard IM abbreviations in the references section."

Query reply 9: Respected reviewer, all references were thoroughly checked and we have incorporated the suggested changes (standard abbreviations) in the revised version of the manuscript.

At the end, we all authors are very thankful for your time and efforts paid to review the resubmitted manuscript. Indeed, your inputs contributed significant improvements. We welcome any other suggestion.

Reviewer 2 Report

- In this manuscript as a whole: Caution should be used when using the terms such as `association` and `associated` in this study, since this study used a cross-sectional study design. Use the appropriate terminology in line with the methodology applied in this study.  

- Lines 57-59: The appropriate reference was not cited. Instead of the reference No. 1, cite the original appropriate reference.     - Lines 59-60: The appropriate reference was not cited. Instead of the reference No. 2, cite the original appropriate reference.    - Text of this manuscript as a whole: Check all references, and cite correctly.      - Line 87: Add a new paragraph that will in detail state all indications for vaccination against COVID-19 in Sudan, whether it is mandatory (and for which indications), whether it is recommended (and for which indications), or not.     - Line 226: For the booster dose for the question `Do you have allergy with any vaccine?` statistical significance was reached (p=0.03). State this in this paragraph.     - Lines 242-243: Add allergy.     - 264-266: Since statistical significance was not reached for any of the vaccine types in this study, one should be cautious when making the following statement `Taken together, the high prevalence of side effects in our study can be linked to the type of vaccine.`. Rephrase or delete this sentence.     - Line 326: Add a new paragraph that will discuss the statistically significant difference regarding the marital status for first and second dose.    - Line 326: Add a new paragraph that will discuss the statistically significant difference regarding the question `Do you have allergy with any vaccine?` for booster dose.     - Line 363: The information for allergy is missing.  

Author Response

Point-by-Point response to reviewers

Respected Editor,

We have received comments for our manuscript. All the concerns and suggestions of the reviewers have been addressed and we hope the revised version of the manuscript will satisfy the concerns of all reviewers. We have attached a point-by-point response to each reviewer and revised version of the manuscript for your consideration. Please let us know if any other changes are required in this regard. Last but not least, we are very thankful to the editor and all reviewers for their time and efforts to put valuable suggestions. Indeed, their recommendations made this manuscript more scientifically elegant and sound.

Query 1: - In this manuscript as a whole: Caution should be used when using the terms such as `association` and `associated` in this study, since this study used a cross-sectional study design. Use the appropriate terminology in line with the methodology applied in this study. 

Response Query 1: Respected Reviewer, we agree with your suggestion. We have rephased the sentences where the word association or associated was present. We hope that the revised version will satisfy the concern.

Query 2: - Lines 57-59: The appropriate reference was not cited. Instead of the reference No. 1, cite the original appropriate reference.    

Response Query 2: Respected reviewer, the reference of the WHO has been cited now as per suggestion.

Query 3: - Lines 59-60: The appropriate reference was not cited. Instead of the reference No. 2, cite the original appropriate reference.   

Response Query 3: Respected Reviewer, the reference of the WHO has been cited now as per suggestion.

Query 4: -Text of this manuscript as a whole: Check all references, and cite correctly.    

Response Query 4: Respected Reviewer, the other references have also been checked and corrected, where necessary.

Query 5: - Line 87: Add a new paragraph that will in detail state all indications for vaccination against COVID-19 in Sudan, whether it is mandatory (and for which indications), whether it is recommended (and for which indications), or not.    

Response Query 5: Respected Reviewer, the COVID-19 vaccination is not mandatory in Sudan. We did not come across any advisory or policy notes from the health ministry where these vaccines are confirmed as mandatory. However, the population of Sudan is encouraged to take the doses through electronic or paper media. However, we have provided this information in the introduction section of the manuscript (3rd Paragraph)

Query 6: - Line 226: For the booster dose for the question `Do you have allergy with any vaccine? ` statistical significance was reached (p=0.03). State this in this paragraph.    

Response Query 6: Respected Reviewer, the suggested sentences have been added in this paragraph.

Query 7: - Lines 242-243: Add allergy.    

Response Query 7: Respected Reviewer, the variable “allergy” has been added at the suggested place.

Query 8: - 264-266: Since statistical significance was not reached for any of the vaccine types in this study, one should be cautious when making the following statement `Taken together, the high prevalence of side effects in our study can be linked to the type of vaccine.`. Rephrase or delete this sentence.    

Response Query 8: Respected Reviewer, we have rephrased the sentence as per suggestion. However, we would like to clarify that purpose of this statement was to discuss about the high prevalence of side effects in our study as compared to other studies. The previous sentences discussed about the variations in prevalence of side effects after the first and second dose. We explained that this variation might be associated with the use of different vaccines. As AstraZeneca was most commonly used vaccine in our study, and this vaccine has already been reported to cause more side effects as compared to other vaccines. We have discussed the same here. However, based on your suggestion we have rephrased the sentence to avoid any confusion among the readers. We hope that revised manuscript will satisfy the concern.

Query 9: - Line 326: Add a new paragraph that will discuss the statistically significant difference regarding the marital status for first and second dose.   

Response Query 9: Respected Reviewer, we have added one paragraph in which we have discussed about the significantly difference of side effect scores according to the marital status of study participants. I hope this information will satisfy the concern.

Query 10: - Line 326: Add a new paragraph that will discuss the statistically significant difference regarding the question `Do you have allergy with any vaccine?` for booster dose.    

Response Query 10: Respected Reviewer, we have added one paragraph in which we have discussed about the significantly difference of side effect scores according to the allergy status of the study participants. I hope this information will satisfy the concern.

Query 11: - Line 363: The information for allergy is missing. 

Response Query 11: Respected Reviewer, the allergy information has been added.

At the end, we all authors are very thankful for your time and efforts paid to review the resubmitted manuscript. Indeed, your inputs contributed significant improvements. We welcome any other suggestion.

Reviewer 3 Report

The manuscript “A First Report on Side Effects of COVID-19 Vaccines among General Population in Sudan: A Cross-sectional analysis” assessed prevalence of COVID-19 vaccine-related side effects after primary and booster doses. The study is important as the first study to assess safety profile of COVID-19 vaccines in Sudan population.

The work is interesting and precedent, although it is preliminary.

I think it is acceptable after some revision, taking into account the following points.

Minor points:

1.    Table 1, 3, 5, 7, and 8 are not referred in the main text. Please refer these tables in the proper section of the text.

Author Response

Point-by-Point response to reviewers

Respected Editor,

We have received comments for our manuscript. All the concerns and suggestions of the reviewers have been addressed and we hope the revised version of the manuscript will satisfy the concerns of all reviewers. We have attached a point-by-point response to each reviewer and revised version of the manuscript for your consideration. Please let us know if any other changes are required in this regard. Last but not least, we are very thankful to the editor and all reviewers for their time and efforts to put valuable suggestions. Indeed, their recommendations made this manuscript more scientifically elegant and sound.

Query 1: The manuscript “A First Report on Side Effects of COVID-19 Vaccines among General Population in Sudan: A Cross-sectional analysis” assessed prevalence of COVID-19 vaccine-related side effects after primary and booster doses. The study is important as the first study to assess safety profile of COVID-19 vaccines in Sudan population.

The work is interesting and precedent, although it is preliminary.

I think it is acceptable after some revision, taking into account the following points.

Minor points:

  1. Table 1, 3, 5, 7, and 8 are not referred in the main text. Please refer these tables in the proper section of the text.

Query reply 1: Respected reviewer, thank you very much for acknowledging our work. Considering the need of such studies in Sudan, we tried our best to provide a real-world data on the vaccine safety in the country. We have incorporated the suggested changes and have cited the tables at appropriate places in text. Please accept our apology for this inconvenience.

At the end, we all authors are very thankful for your time and efforts paid to review the resubmitted manuscript. Indeed, your inputs contributed significant improvements. We welcome any other suggestion.
